# Prevalence and determinants of hyperglycaemia among adults in Bangladesh: results from a population-based national survey

Jessica Yasmine Islam,[1,2] Mohammad Mostafa Zaman,[1]
Mahfuz Rahman Bhuiyan,[1] Syed Atiqul Haq,[3] Shamim Ahmed,[3]
Ahmad Zahid Al-Qadir[3]

¹Non-Communicable Disease Unit, World Health Organization Country Office for Bangladesh, Dhaka, Bangladesh
²Department of Epidemiology, University of North Carolina at Chapel Hill Gillings School of Global Public Health, Chapel Hill, North Carolina, USA
³Rheumatology, Bangabandhu Sheikh Mujib Medical University, Dhaka, Bangladesh

**Correspondence to**
Dr Mohammad Mostafa Zaman;
zamanm@who.int

## ABSTRACT

**Objectives** With the increasing burden of non-communicable diseases in low-income and middle-income countries, biological risk factors, such as hyperglycaemia, are a major public health concern in Bangladesh. Hyperglycaemia is an excess of glucose in the bloodstream and is often associated with type 2 diabetes mellitus. Nationally representative data of hyperglycaemia prevalence starting from age ≥18 years are currently unavailable for Bangladeshi adults. The objective of this study was to assess the prevalence and determinants of hyperglycaemia among adults in Bangladesh aged ≥18 years.

**Study design** Cross-sectional, population-based study.

**Setting and participants** Data for this analysis were collected in November to December 2015, from a population-based nationally representative sample of 1843 adults, aged ≥18 years, from both urban and rural areas of Bangladesh. Demographic information, capillary blood glucose, blood pressure, height, weight, waist circumference and treatment history were recorded.

**Primary outcome measures** Hyperglycaemia was defined as a random capillary blood glucose level of ≥11.1 mmol/L (ie, in the diabetic range) or currently taking medication to control type 2 diabetes, based on self-report.

**Results** Overall, the prevalence of hyperglycaemia was 5.5% (95% CI 4.5% to 6.6%) and was significantly higher among urban (9.8%, 95% CI 7.7% to 12.2%) than rural residents (2.8%, 95% CI 1.9% to 3.9%). The age-standardised prevalence of hyperglycaemia was 5.6% (95% CI 4.6% to 6.8%). Among both urban and rural residents, the associated determinants of hyperglycaemia included hypertension and abdominal obesity. About 5% of the total population self-reported have been previously diagnosed with type 2 diabetes; among these adults, over 25% were not taking medications to control their diabetes.

**Conclusions** Our study found that about 1 in 20 Bangladeshi adults aged ≥18 years have hyperglycaemia. To control and prevent the development of type 2 diabetes, data from this study can be used to inform public health programming and provide descriptive information on surveillance of progress towards controlling diabetes in Bangladesh.

## Strengths and limitations of this study

► This study used a multistage, geographically clustered, probability-based sampling approach to produce nationally representative data for Bangladesh.

► Currently, nationally representative data for the prevalence of hyperglycaemia in the diabetic range are unavailable for adults aged 18–29 years. A strength of our study is that we included Bangladeshi adults aged 18 years and above to obtain novel data on the prevalence of hyperglycaemia and relevant non-communicable disease risk factors.

► We were able to estimate the prevalence of hyperglycaemia using capillary blood glucose measured at random. However, we were unable to measure the prevalence of pre-diabetes and diabetes as we did not obtain blood sugar levels using standardised methods, such as fasting blood glucose or 2-hour postprandial measurements.

► Due to the cross-sectional nature of the study design, we were unable to assess temporality of risk factors identified and our outcomes of interest.

► We assessed type 2 diabetes medication history based on self-report and we were able to obtain medicine strips or vials records or prescription records of participants to confirm the self-reported data.

## BACKGROUND

Globally, diabetes mellitus (DM), characterised by hyperglycaemia or high blood glucose, is a leading cause of premature mortality and disability. Globally, almost half of all deaths attributable to high blood glucose occur before the age of 70 years. Worldwide, the prevalence of DM has been on the rise over the past several decades.[1 2] In fact, estimates reflect that the global prevalence of diabetes has nearly doubled among adults aged 18 years and above, rising from 4.7% in 1980 to 8.5% in 2014.[3] This growing burden is most prominent in low-income and middle-income

countries (LMICs) particularly the Indian subcontinent,[4] which accounts for close to one-fifth of all diabetes cases worldwide. The prevalence of diabetes in this region is projected to increase by 71% by 2035.[5] In Bangladesh, specifically, the International Diabetes Federation projects the prevalence of diabetes will increase to more than 50% in the next 15 years[5].

The increasing prevalence of diabetes among Bangladeshi adults over the past few decades has been documented: Based on a meta-analysis of studies conducted from 1995 to 2010, the prevalence of diabetes among Bangladeshi adults aged 30 years and above increased from 4% in 1995 to 2000 to 9% in 2006 to 2010.[6] Studies to assess the burden of diabetes have been conducted in Bangladesh in both urban and rural populations over the last decades.[7] However, national data on the prevalence of diabetes or hyperglycaemia in the diabetic range starting at age 18 years are currently unavailable.[7–10] These data are valuable for monitoring progress made towards one of the nine global non-communicable disease (NCD) targets of the year 2025, set forth by WHO's NCD Global Monitoring Framework: To observe a 0% increase in age-standardised prevalence of hyperglycaemia or diabetes among persons aged ≥18 years.[11] As such in response to WHO Global Action Plan for the Prevention and Control of NCDs,[12] descriptive epidemiological data on the burden of hyperglycaemia among adults starting at 18 years are needed to monitor the national progress of interventions implemented to reduce the burden of DM in Bangladesh.[13–15] Here, using a nationally representative sample, we present data on prevalence and determinants of hyperglycaemia by various sociodemographic factors such as age, sex and area of residence among Bangladeshi adults aged 18 years and above, residing in both urban and rural areas of the country. Additionally, we explore treatment patterns and control of hyperglycaemia among participants in our sample of Bangladeshi adults.

## METHODS
Data for this analysis were collected as part of a national assessment of the burden of musculoskeletal disorders in Bangladesh conducted by investigators from the Bangabandhu Sheikh Mujib Medical University with technical assistance from WHO Country Office for Bangladesh, as previously described.[16] The study was a population-based cross-sectional study carried out from November to December 2015 and followed WHO STEPwise approach to Surveillance of NCD risk factors (STEPS).[17] The target population of this survey was men and women aged ≥18 years residing in rural and urban areas of Bangladesh. The exclusion criteria included tourists and the institutionalised, including residents of hospitals, prisons, nursing homes and army barracks.

### Sampling methods
To obtain a population-based sample of Bangladesh, this survey adopted a multistage, geographically clustered,

probability-based sampling approach. Population statistics were obtained using the updated national census conducted by the Bangladesh Bureau of Statistics (BBS) in 2009.[18] To obtain our primary sampling unit (PSU), we used the following geographical distribution described: In Bangladesh, there are seven divisions, which are the largest administrative units of the country. Each division is divided into several districts (zila) and within each district, there are several subdistricts (upazila). Within subdistricts, mauzas and mahallas (commonly known as neighbourhoods or blocks) are the smallest units within defined territories in rural and urban areas, respectively. Mauzas and mahallas were considered the PSU for the study's sampling approach. The households within the mauzas and mahallas were the secondary sampling units. We used the BBS definition of a household, which is as follows: 'a dwelling in which persons either related or unrelated were living together and taking food from the same kitchen'.[18]

### Sample size estimation
The power analysis and sample size calculations were completed based on the standardised approach outlined in WHO STEPS methodology.[17] Using WHO STEPS methodology, the minimum number of participants required was 296 in each group (rural males, rural females, urban males and urban females). Assuming a design effect of 1.5 adjusted within cluster population homogeneity, the necessary sample size was 1776. We assumed a response rate of 90% and determined we would need to contact at least 1973 adults. For simplicity, our target sample size was 2000. Twenty PSUs (8 urban and 12 rural) were randomly selected from seven divisions of the country, with the probability proportional to the population size of each division. In each PSU, 100 consecutive households were selected. The even-numbered households were designated as a 'male household' and odd-numbered households as a 'female household.' Finally, one male or female was approached to participate from each respective household as designated.

### Data collection
Through a structured survey, we collected data on the following topics: musculoskeletal disorders,[19] health history and demographic data, such as age, area of residence, education, current (last 12 months) occupation, tobacco use and physical activity. Physical measurements, such as height, weight, waist circumference, blood glucose levels and blood pressure (BP), were collected. To measure blood glucose levels, we obtained random blood glucose samples,[3] as per the clinical guidelines of diabetes diagnostic criteria of Bangladesh.[20] Capillary blood samples were consistently taken from the index finger of the right arm using a glucometer, namely Accu-Chek Advantage (Roche Diagnostics Division, Grenzacherstrasse, Switzerland).

Each participant's history of diabetes was assessed based on self-report. Specifically, participants were

asked: (1) Have you ever been diagnosed with diabetes by a healthcare professional? (2) If yes, are you receiving treatment for diabetes? Treatment history of diabetes was confirmed by prescription, including medicine strips or insulin injection vials or medical records. The questionnaire was translated from English into Bengali, adapted and validated as per standard procedure. Data collection procedures were standardised across study sites through coordinated training of field staff conducted by epidemiologists, study physicians and WHO staff members.

BP was measured by a trained field interviewer using the LifeSource UA-767 +blood pressure monitor, as recommended by WHO, and appropriately sized arm cuffs. BP measurements were consistently taken on each participant's right arm at the level of the heart and elbow assisted, while the participant was in a seated position. The initial measurement was performed after 5 min of rest. After 2 min, the second measurement was taken. The mean of these two BP readings was used as the final BP for each participant.

Following data collection, participants scheduled a visit with the study research physician within the following 5 days. The research physician assessed the participant's medical history, either through self-report or using medical records when possible. Study physicians examined each participant to confirm the results of data collection through classical symptom assessment. When necessary, the participant was also evaluated by the divisional investigator for a second opinion of relevant diagnoses.

### Patient and public involvement
There was no patient or public involvement in the implementation of this study or interpretation of analytical results.

### Outcome definitions
Our primary outcome of interest was the prevalence of hyperglycaemia. We used the American Diabetes Association (ADA) guidelines to define a diagnosis of hyperglycaemia using random or casual plasma glucose test and symptom review by the study physician.[21] An individual was considered to have hyperglycaemia if the plasma glucose level was 11.1 mmol/L or higher (ie, in the diabetic range) and/or if they self-reported to take diabetes medication. Our secondary outcome of interest was the prevalence of self-reported type-2 DM. An individual was categorised as diabetic if they self-reported to have been previously diagnosed with diabetes by a healthcare provider.

### Covariates
The following variables were assessed as covariates for analysis: area of residence, sex, age, education, occupation, wealth index, body mass index (BMI), BP and waist circumference. Education was categorised into four groups: no education, primary education (completed ≤grade 5), secondary education (completed ≤grade 10) and above secondary education (completed ≥grade 12). Each participant's occupation was categorised into five groups, including: professional employment (field staff, police officer, guard, doctor, engineer, professional, businessman and desk job), unemployed or retired, industrial worker or day labourer, housewife and other (shop keeper, weaver, driver, student, beggar, cook, carpenter, tailor, migrant workers and fishermen).

Data on physical activity were collected based on self-report. First, respondents were asked the number of days they engaged in vigorous, moderate or light physical activity throughout a typical week. The following definitions were used to define (1) vigorous, (2) moderate and (3) light physical activity, respectively: (1) vigorous activity was defined as any activity that caused a large increase in breathing or heart rate, if continued for at least 10 min (eg, running, carrying heavy loads, digging or construction work); (2) moderate activity was defined as any activity that caused a small increase in breathing or heart rate, if continued for at least 10 min (brisk walking or carrying light loads) and (3) light physical activity was defined as activities, such as office work. Next, we asked participants to estimate how many minutes per day they engaged in the activity. Metabolic equivalent of task (MET) minute was calculated using the STEPS protocol[22] as follows: 1 min of light activity was equivalent to 1 MET-minute; 1 min in moderate-intensity activities was equivalent to 4 MET-minutes, and 1 min of vigorous-intensity was equivalent to 8 MET-minutes. Physical activity was then categorised based on the total MET-minutes per week. Participants who spent 3000 or more MET-minutes per week were categorised in the vigorous physical activity group, 600–3000 MET-minutes were categorised as moderate physical activity and <600 MET-minutes were categorised as low physical activity.

The wealth index was constructed using principal component analysis. Asset information collected covered information on household ownership of nineteen items, including electricity, flush toilet, land telephone, cell phone, television, radio, refrigerator, car, motorcycle, washing machine, bicycle, sewing machine, wardrobe, table, bed or cot, chair or bench, watch or clock. Additionally, we assessed the main type of material used to build each participant's home (ie, cement, tin, bamboo or thatched straw). Each asset was assigned a weight (factor score) generated through principal component analysis, and the resulting asset scores were standardised to a normal distribution with a mean of 0 and SD of 1. Each household was then assigned a score for each asset, and the scores were summed up; individuals were ranked according to the total score of the household in which they resided. The sample was then divided into quartiles from quartile one (lowest) to quartile four (highest).

Using height (centimetres) and weight (kilograms) measurements, we calculated BMI (weight/height$^2$). BMI was categorised into the following groups: underweight (≤18.5 kg/m$^2$), normal (≤25 kg/m$^2$), overweight (25.1–30 kg/m$^2$) and obese (>30 kg/m$^2$). Waist

circumference was measured in centimetres. Participants were categorised as abdominally obese if waist circumference was 90 cm and above for men, or 80 cm and above for women. Prehypertension was defined as systolic BP (SBP) ≥120 mm Hg but <140 mm Hg and/or diastolic BP (DBP) ≥80 mm Hg but <90 mm Hg and not taking antihypertensive medication at the time of the survey. We used WHO's guidelines for cut-off points to define hypertension.[23] An individual was considered to have hypertension if SBP was ≥140 mm Hg (millimetres of mercury) and/or, DBP ≥90 mm Hg, and/or taking antihypertensive medication based on self-report.

### Age-standardised prevalence estimates

To facilitate comparison of overall hyperglycaemia among Bangladeshi adults across global populations with different age compositions, we calculated age-standardised prevalence estimates with 95% CIs using WHO's World Standard Population.[24] The World Standards database (WHO 2000–2025) provided population estimates for 18 and 19 age groups, as well as single year ages. To derive single ages from the 5-year age group proportions publicly available, we used the Beers 'Ordinary' Formula.[25] The following formula was used for standardisation:

$\sum p_i * w_i / \sum w_i$, where p=observed prevalence and w=world population weight.

### Data analysis

Sociodemographic variables were presented with mean and SD for continuous variables, and using proportions for categorical variables. For bivariate analyses, study participants were divided by sex and into four age groups (18–29, 30–44, 45–54 and ≥55 years). We calculated the prevalence of our primary outcome by key demographic variables and calculated 95% CIs using the binomial exact method.

To estimate determinants of hyperglycaemia in the diabetic range, we computed prevalence ratios with Poisson regression using robust estimation of standard errors.[26–28] Potential variables for inclusion in the model were assessed using the prior published literature and bivariate Poisson regression analysis; an arbitrary p<0.10 was used as criteria to include the variable in the multivariable Poisson regression model to control for confounding effects. For multivariable Poisson regression models, adjusted prevalence ratios (aPRs) and 95% CIs for each independent variable were calculated. Additionally, <0.05 was used as the level of significance. Multivariable Poisson regression models were generated separately for urban and rural participants to account for possible effect measure modification. Collinearity was assessed using the variance inflation factor to ensure a strong linear relationship among independent variables included in the model was not present. All statistical procedures were performed using Stata/SE V.15.0 (StataCorp LP) software package.

## RESULTS

### Background characteristics

Of the 2000 adults approached, 1843 agreed to participate in our study leading to a response rate of 92.1%. Twenty-four female participants were pregnant and were dropped from subsequent analyses to ensure those with gestational diabetes were not included. Additionally, no participants reported having been previously diagnosed with type-1 diabetes. There were 892 (49.0%) male and 927 (50.9%) female respondents (table 1). The age of our participants ranged from 18 to 90 years. The mean age and education level of participants were 40.5 (SD=14.7) years and 5.7 (SD=5.1) years, respectively. The majority of the study population was married (88.0%) and employed as either an industrial worker/day labourer (26.5%) or housewife (39.9%). Almost half of participants never used some form of tobacco. The majority of participants engaged in vigorous physical activity over an average week (69.3%). The mean BMI was 22.1 (SD=4.1) and the mean waist circumference was 78.4 cm (SD=11.6). Overall, the mean SBP was 116.1 mm Hg (SD=17.1) and DBP was 76.1 mm Hg (SD=10.5). The mean blood glucose level was 6.4 mmol/L (SD=2.4). Figure 1 presents the distribution of blood glucose levels by various demographic factors.

### Prevalence and risk factors for hyperglycaemia

The prevalence of hyperglycaemia was 5.5% (95% CI 4.5% to 6.6%). This prevalence was significantly higher among urban participants (9.8%, 95% CI 7.7% to 12.2%) than rural participants (2.8%, 95% CI 1.9% to 3.9%) (table 2) and increased as age increased (figure 2). The highest prevalence of hyperglycaemia was observed among those aged ≥55 years, at 8.2% (95% CI 4.6% to 13.1%) among men and 9.4% (95% CI 5.4% to 14.8%) among women. The age-standardised prevalence of hyperglycaemia was 5.6% (95% CI 4.6% to 6.8%). The age-standardised prevalence of hyperglycaemia among urban and rural residents was 10.5% (95% CI 9.2% to 12.1%) and 2.8% (95% CI 2.1% to 3.7%), respectively. Among men and women, the age-standardised prevalence of hyperglycaemia was 4.9% (95% CI 3.9% to 6.0%) and 6.0% (95% CI 4.9% to 7.2%), respectively.

### Self-reported DM and hyperglycaemia

Ninety-five participants (5.2%) self-reported to have been previously diagnosed with type 2 diabetes by a healthcare provider. However, 25 of those participants did not meet our criteria of diagnosis of hyperglycaemia as they did not take medication to control their diabetes and their plasma glucose was below 11.1 mmol/L. Therefore, 69.3% of those with hyperglycaemia, as per the study definition, were previously diagnosed with diabetes by a healthcare provider.

The proportion of men and women, who were previously diagnosed with diabetes by a healthcare provider based on self-report, was higher among urban residents (men: 8.4%; women: 9.1%) than rural residents (men: 2.9%; women: 3.0%) (figure 3A). However, overall, the

**Table 1** Background characteristics of Bangladeshi adult participants, 2015 (n=1819)

| Characteristic | Total (n=1819) | | | Urban (n=708) | | | Rural (n=1111) | | |
|---|---|---|---|---|---|---|---|---|---|
| | Mean (SD) | n | % | Mean (SD) | n | % | Mean (SD) | n | % |
| Sex | | | | | | | | | |
| Male | | 892 | 49.0 | | 345 | 48.7 | | 547 | 49.2 |
| Female | | 927 | 50.9 | | 363 | 51.3 | | 564 | 50.8 |
| Age (years) | 40.5 (14.7) | | | 39.1 (13.9) | | | 41.4 (15.1) | | |
| Education (years)* | 5 (0–9) | | | 8 (3–12) | | | 4 (0–8) | | |
| Marital status | | | | | | | | | |
| Never married | | 110 | 6.1 | | 54 | 7.6 | | 56 | 4.9 |
| Married | | 1601 | 88.0 | | 619 | 87.4 | | 982 | 88.4 |
| Separated/divorced/ widowed | | 108 | 5.9 | | 35 | 4.9 | | 73 | 6.5 |
| Occupation | | | | | | | | | |
| Professional employment† | | 279 | 15.2 | | 189 | 26.7 | | 90 | 8.1 |
| Unemployed/retired | | 98 | 5.3 | | 43 | 6.1 | | 55 | 4.9 |
| Industrial worker/day labourer | | 483 | 26.6 | | 120 | 16.9 | | 363 | 32.7 |
| Housewife | | 726 | 39.9 | | 247 | 34.9 | | 479 | 43.2 |
| Other‡ | | 232 | 12.8 | | 109 | 15.4 | | 123 | 11.1 |
| Wealth Index§ | | | | | | | | | |
| 1st wealth quartile | | 407 | 22.4 | | 110 | 15.5 | | 297 | 26.7 |
| 2nd wealth quartile | | 533 | 29.3 | | 171 | 24.2 | | 362 | 32.6 |
| 3rd wealth quartile | | 429 | 23.6 | | 179 | 25.3 | | 250 | 22.5 |
| 4thourth Wealth quartile | | 450 | 24.7 | | 248 | 35.0 | | 202 | 18.2 |
| Tobacco use¶ | | | | | | | | | |
| Never | | 859 | 47.2 | | 389 | 54.9 | | 470 | 42.3 |
| Current use | | 821 | 45.1 | | 268 | 37.9 | | 553 | 49.8 |
| Past use | | 139 | 7.6 | | 51 | 7.2 | | 88 | 7.9 |
| Smoking tobacco use** | | | | | | | | | |
| Every day/occasionally | | 494 | 27.2 | | 169 | 23.9 | | 325 | 29.3 |
| Past use | | 104 | 5.7 | | 38 | 5.4 | | 66 | 5.9 |
| Never | | 1221 | 67.1 | | 501 | 70.8 | | 720 | 64.8 |
| Smokeless tobacco use†† | | | | | | | | | |
| Every day/occasionally | | 529 | 29.1 | | 150 | 21.2 | | 379 | 34.1 |
| Past use | | 51 | 2.8 | | 21 | 2.9 | | 30 | 2.7 |
| Never | | 1239 | 68.1 | | 537 | 75.9 | | 702 | 63.2 |
| Physical activity‡‡ | | | | | | | | | |
| Vigorous | | 1268 | 69.7 | | 445 | 62.9 | | 823 | 74.1 |
| Moderate | | 464 | 25.5 | | 234 | 33.1 | | 230 | 20.7 |
| Low | | 87 | 4.7 | | 29 | 4.1 | | 58 | 5.2 |
| BMI§§ | 22.1 (4.1) | | | 23.3 (4.5) | | | 21.3 (3.7) | | |
| Waist circumference (cm) | 78.4 (11.6) | | | 81.8 (12.6) | | | 76.2 (10.3) | | |
| Blood pressure | | | | | | | | | |
| Systolic blood pressure (mm Hg) | 116.1 (17.1) | | | 117.9 (16.6) | | | 115.0 (17.3) | | |

**Table 1** Continued

| Characteristic | Total (n=1819) | | | Urban (n=708) | | | Rural (n=1111) | | |
|---|---|---|---|---|---|---|---|---|---|
| | Mean (SD) | n | % | Mean (SD) | n | % | Mean (SD) | n | % |
| Diastolic blood pressure (mm Hg) | 76.1 (10.5) | | | 77.9 (10.9) | | | 74.9 (10.1) | | |
| Blood glucose level (mmol/L) | 6.4 (2.4) | | | 6.6 (2.9) | | | 6.3 (2.1) | | |
| Self-reported diabetes medication history¶¶ | | 69 | 72.6 | | 50 | 80.6 | | 19 | 57.6 |

*Calculated median and IQR for education as the data are skewed.
†Professional occupation includes: field staff, police officer, guard, doctor, engineer, professional, business man, desk job.
‡Other occupation includes: shop keeper, weavers, driver, student, beggar, cook, carpenter, tailor, migrant workers and fishermen.
§Wealth index was calculated using principal component analysis using data collected on household ownership of the following items: electricity, flushable toilet, land phone, cell phone, television, radio, refrigerator, private car, motor cycle, washing machine, bicycle, sewing machine, almirah/wardrobe, table, bed, chair/bench, watch/clock, as well as, type of main material used to build their homes roof, walls and floor.
¶Includes both smokeless tobacco and smoke tobacco.
**Smoking tobacco use includes cigarettes, bidi, hookah.
††Smokeless tobacco use includes jarda, sada pata, pan masala with tobacco leaf, gul.
‡‡Measured in MET-minutes; 1 MET stands for the amount of oxygen you consume and the number of calories you burn at rest.
§§BMI calculated by weight in kilogram divided by height in metre squared.
¶¶Percentage reported out of participants who self-reported to have diabetes (total n=95; urban n=62; rural n=33).
BMI, body mass index; MET, metabolic equivalent of task.

large majority (81.8%) reported that they did not know if they had been previously diagnosed with diabetes; this proportion was higher among rural residents (87.8%) than urban residents (72.2%).

Among participants previously diagnosed with type 2 diabetes based on self-report, 72.6% reported taking medication to control their diabetes (table 1). Urban women more frequently (96.7%) self-reported to take diabetes medication than urban men (62.1%) (figure 3B). We were able to confirm 100% of participant's self-reported diabetes treatment history by checking prescriptions or medicine strips/vials.

Among participants who were categorised as hyperglycaemic during study measurement, over one-third (37.9%) of the urban men self-reported to have diabetes, however, they did not take any medication to control their diabetes. Among rural participants, the proportion of women who did not take medication to control their self-reported diabetes was higher (52.9%) than men (31.3%). Although three-quarters of self-reported diabetic participants reported taking medication to control their diabetes, 31% continued to have high blood sugar levels indicating uncontrolled diabetes at study measurement (figure 4).

### Determinants of hyperglycaemia

Table 2 presents the results of multivariable Poisson regression with robust variance analyses to identify determinants

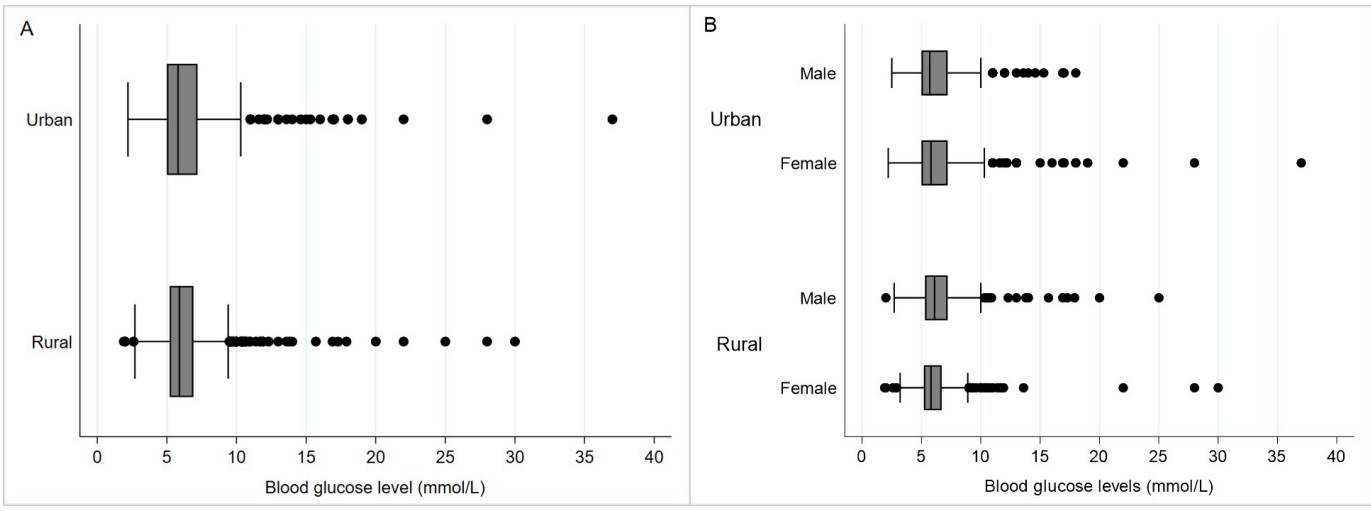

**Figure 1** Urban and rural differences in the distribution of blood glucose levels based on random capillary blood measurement among (A) all participants and (B) men and women (n=1819).

**Table 2**  Determinants of hyperglycaemia among Bangladeshi adults, 2015 (n=1819)

| Characteristic | Total (n=1819) | | Urban (n=708) | | Rural (n=1111) | |
|---|---|---|---|---|---|---|
| | Diabetes* Prevalence % | Adjusted PR† (95% CI) | Diabetes* Prevalence % | Adjusted PR† (95% CI) | Diabetes* Prevalence % | Adjusted PR† (95% CI) |
| **Area** | | | | | | |
| Urban | 9.8 | Ref. | – | – | – | – |
| Rural | 2.8 | 0.44 (0.28 to 0.68) | – | – | – | – |
| **Sex** | | | | | | |
| Male | 4.9 | Ref. | 7.5 | Ref. | 3.3 | Ref. |
| Female | 5.9 | 1.05 (0.71 to 1.54) | 11.9 | 1.26 (0.80 to 1.99) | 2.2 | 0.61 (0.29 to 1.28) |
| **Age (years)** | | | | | | |
| 18–29 | 2.5 | Ref. | 2.5 | Ref. | 2.5 | Ref. |
| 30–44 | 4.9 | 1.48 (0.78 to 2.79) | 8.4 | 2.55 (1.01 to 6.41) | 2.5 | 0.76 (0.30 to 1.93) |
| 45–54 | 7.4 | 2.18 (1.10 to 4.31) | 15.5 | 4.38 (1.62 to 11.59) | 2.8 | 0.67 (0.25 to 1.83) |
| ≥55 | 8.8 | 1.92 (0.95 to 3.86) | 19.3 | 3.92 (1.48 to 10.39) | 3.4 | 0.53 (0.17 to 1.67) |
| **Educational status** | | | | | | |
| No education | 3.4 | Ref. | 7.0 | Ref. | 2.1 | Ref. |
| Primary education | 3.7 | 1.00 (0.53 to 1.86) | 7.6 | 1.00 (0.44 to 2.30) | 2.1 | 0.84 (0.30 to 2.30) |
| Secondary education | 6.9 | 1.67 (0.95 to 2.93) | 12.1 | 1.94 (0.95 to 3.97) | 2.9 | 1.01 (0.40 to 2.54) |
| Above secondary education | 9.8 | 1.48 (0.77 to 2.84) | 10.4 | 1.54 (0.71 to 3.33) | 8.1 | 2.24 (0.68 to 7.41) |
| **Wealth Index‡** | | | | | | |
| 1st wealth quartile | 7.7 | 2.58 (1.57 to 4.24) | 18.9 | 3.18 (1.80 to 5.62) | 3.6 | 1.37 (0.53 to 3.49) |
| 2nd wealth quartile | 4.3 | 1.23 (0.71 to 2.14) | 9.4 | 1.50 (0.82 to 2.77) | 1.9 | 0.70 (0.24 to 2.05) |
| 3rd wealth quartile | 3.7 | 0.86 (0.47 to 1.58) | 6.0 | 0.96 (0.48 to 1.92) | 2.0 | 0.55 (0.18 to 1.73) |
| 4th wealth quartile | 6.6 | Ref. | 8.8 | Ref. | 3.9 | Ref. |
| **Blood pressure** | | | | | | |
| Normal blood pressure | 1.9 | Ref. | 3.5 | Ref. | 1.1 | Ref. |
| Prehypertension§ | 5.7 | 1.74 (1.00 to 3.01) | 8.3 | 1.37 (0.69 to 2.74) | 3.8 | 2.32 (0.91 to 5.92) |
| Hypertension¶ | 18.9 | 3.57 (2.01 to 6.34) | 27.4 | 2.65 (1.30 to 5.38) | 8.8 | 5.39 (1.94 to 14.96) |
| **Physical activity** | | | | | | |
| Vigorous | 3.9 | Ref. | 6.9 | Ref. | 2.3 | Ref. |
| Moderate | 8.2 | 1.18 (0.78 to 1.77) | 13.8 | 1.22 (0.77 to 1.93) | 2.1 | 0.65 (0.22 to 1.88) |
| Low | 14.9 | 3.04 (1.69 to 5.47) | 20.7 | 3.01 (1.42 to 6.38) | 12.1 | 2.58 (0.95 to 7.04) |
| **Body mass index (BMI, kg/m²)**\*\* | | | | | | |
| Underweight (<18.5) | 1.1 | 0.37 (0.12 to 1.13) | 1.8 | 0.38 (0.07 to 1.95) | 0.7 | 0.42 (0.09 to 2.02) |
| Normal (18.5–25) | 4.7 | Ref. | 8.2 | Ref. | 2.8 | Ref. |
| Overweight (25.1–30) | 10.4 | 1.06 (0.68 to 1.65) | 14.0 | 1.05 (0.61 to 1.80) | 6.2 | 1.22 (0.51 to 2.91) |
| Obese (>30) | 20.9 | 1.49 (0.87 to 2.57) | 26.5 | 1.46 (0.81 to 2.64) | 5.6 | 0.95 (0.12 to 7.60) |
| **Waist circumference (cm)** | | | | | | |
| Normal†† | 2.3 | Ref. | 3.8 | Ref. | 1.6 | Ref. |
| Abdominally obese‡‡ | 13.3 | 2.49 (1.53 to 4.07) | 18.1 | 2.54 (1.35 to 4.77) | 6.8 | 2.95 (1.32 to 6.58) |

*Hyperglycaemia was defined as a capillary blood glucose level greater than or equal to 11.1 mmol/L or self-reported diabetes medication use.
†Model adjusted for all variables included in table: sex, age, education, wealth index, blood pressure, body mass index, self-reported physical activity and waist circumference.
‡Wealth index was calculated using principal component analysis using data collected on household ownership of the following items: electricity, flushable toilet, land phone, cell phone, television, radio, refrigerator, private car, motor cycle, washing machine, bicycle, sewing machine, almirah/wardrobe, table, bed, chair/bench, watch/clock, as well as, type of main material used to build their homes roof, walls and floor.
§Prehypertension was defined as SBP ≥120 mm Hg but <140 mm Hg and/or DBP ≥80 mm Hg but <90 mm Hg and not taking antihypertensive medication at the time of the survey.
¶Hypertension was defined as SBP was ≥140 mm Hg (millimetres of mercury) and/or, DBP ≥90 mm Hg and/or taking antihypertensive medication.
\*\*BMI calculated by weight in kilogram divided by height in metre squared.
††Defined as <90 cm M; <80 cm F.
‡‡Defined as ≥90 cm M; ≥80 cm F.
DBP, diastolic blood pressure; PR, prevalence ratio; Ref, referent category; SBP, systolic blood pressure.

of hyperglycaemia. Among urban participants, those of older age, lowest wealth quartile, hypertension, low physical activity and with abdominal obesity based on waist circumference, were more likely to have hyperglycaemia. The prevalence of hyperglycemia was significantly highest among those aged ≥55 years (aPR 3.92, 95% CI 1.48 to 10.39) compared with individuals aged 18–29 years of age. When compared with those in the fourth (highest) wealth quartile, urban residents in the first (lowest) wealth quartile had 3.18 times the prevalence of hyperglycaemia.

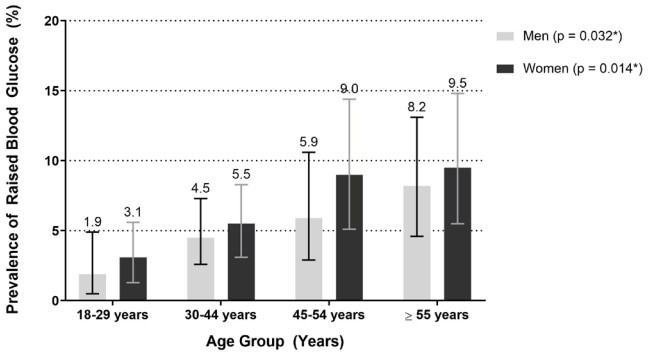

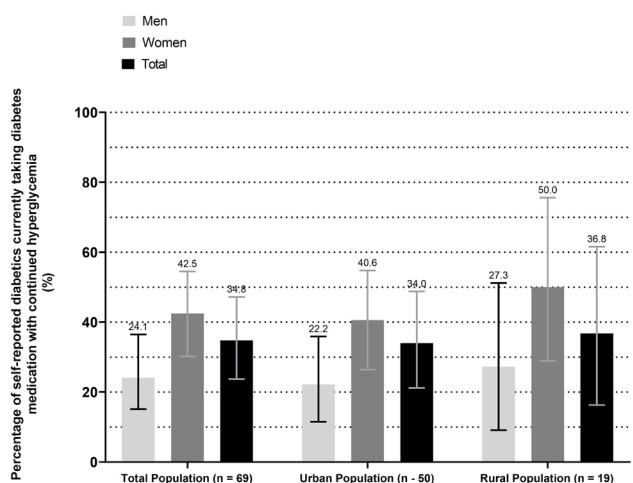

**Figure 2** Prevalence of hyperglycaemia among Bangladeshi adults aged 18 years and above by sex and age group, 2015 (n=1819).

**Figure 4** Self-reported diabetics aged 18 years or older who take diabetes medication with hyperglycaemia on study measurement (≥11.0 mmol/L).

For urban individuals with hypertension, the prevalence of hyperglycaemia was 2.65 (95% CI 1.30 to 5.38) times that of individuals without hypertension. The prevalence of hyperglycaemia among those with low physical activity was 3.01 (95% CI 1.42 to 6.38) times that of urban participants with vigorous physical activity. Abdominal obesity also significantly increased the prevalence of hyperglycaemia among urban participants (aPR 2.54, 95% CI 1.35 to 4.77). Among rural participants, the only observed determinants of hyperglycaemia were hypertension (aPR

5.39, 95% CI 1.94 to 14.96) and abdominal obesity (aPR 2.95, 95% CI 1.32 to 6.58).

## DISCUSSION

Using data from this nationally representative sample, we estimate that about 1 in 20 Bangladeshi adults aged ≥18 years have hyperglycaemia. The prevalence

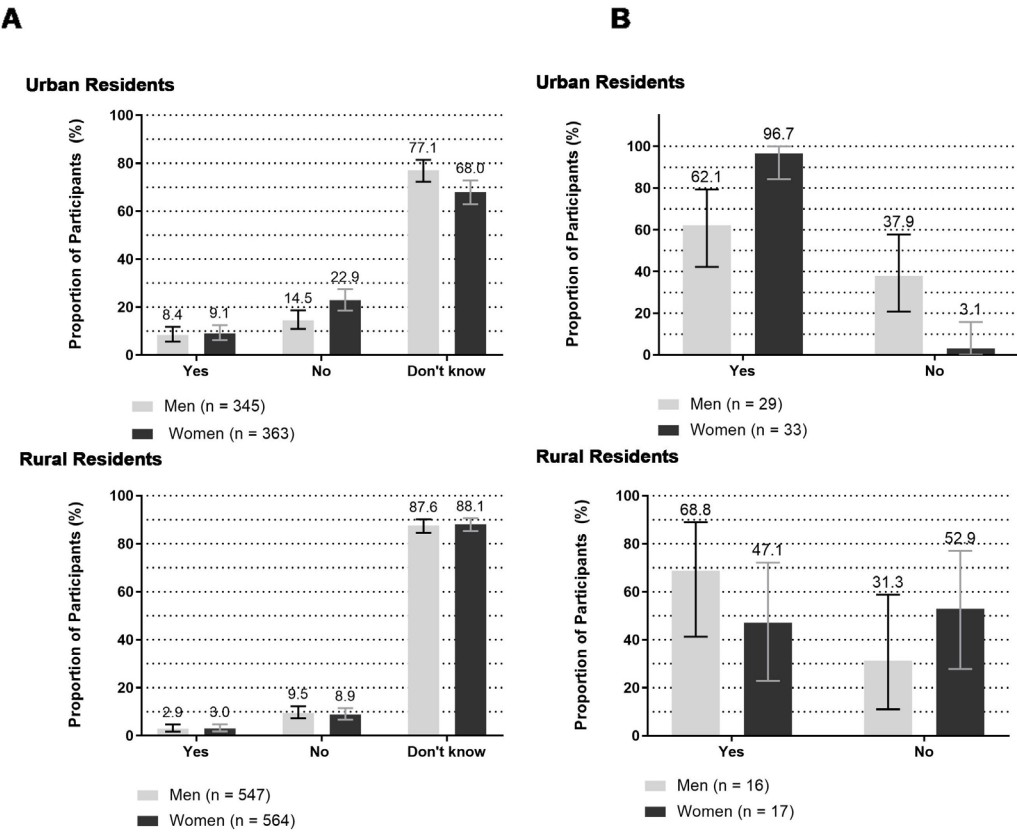

**Figure 3** Bangladeshi adults aged 18 years and above with (A) self-reported diabetes and (B) self-reported diabetics on diabetes medications, 2015.

of hyperglycaemia was higher among urban residents (9.8%) than rural residents (2.8%). Determinants of hyperglycaemia included older age, urban residence, abdominal obesity, low physical activity and hypertension. As diabetes is characterised by hyperglycaemia, targeting high-risk groups identified in this analysis could be prioritised for effective diabetes preventive programmes in Bangladesh.

Bangladesh has adopted the goals and targets set forth by WHO's Global Monitoring Framework for the Prevention and Control of NCDs (2010-2025).[11] One of these targets is to ensure there is a 0% increase in the age-standardised prevalence of hyperglycaemia among adults aged ≥18 years by the year 2025. To our knowledge, the present study is the first to report national estimates on the prevalence of hyperglycaemia starting at age 18 years in both urban and rural areas of Bangladesh. Data gathered from this national-level study are critical towards the measurement of progress towards WHO's nine global NCD control targets for Bangladesh for 2025.

Globally, the number of adults living with diabetes has risen from 108 million in 1980 to 422 million in 2013, and LMICs have seen the most rapid rise in diabetes prevalence.[3] Several lifestyle factors have been attributed to the increase in prevalence across LMICs including, globalisation of food production, extensive marketing of low-cost and energy-dense foods, increased sedentary behaviour and rapid urbanisation.[29] In recent decades, the increase in the prevalence of diabetes in South Asia has been greater than that seen in high-income countries.[4] Based on recent estimates, the prevalence of diabetes in adults across countries in South Asia is similar, excluding Nepal which has a low prevalence in comparison to neighbouring countries (8.8% in India, 8.6% in Sri Lanka, 6.9% in Bangladesh, 7.9% in Bhutan, 6.9% in Pakistan and 4.0% in Nepal).[30] In our study, we observed a prevalence of 5.6% hyperglycaemia in the diabetic range among adults aged 18 years and above. This prevalence is lower than previous studies conducted in Bangladesh and neighbouring countries. In fact, a scoping review estimated the pooled prevalence of type 2 diabetes to be 7.4%.[7] Our estimated prevalence was lower due to a younger study population (18 years and above compared with WHO estimate among adults aged 30 years and above), and a higher percentage of participants from rural areas (61.1%, as is representative of Bangladesh). Indeed, when we restrict our analytical sample to 30 years and above, the prevalence of hyperglycaemia is 6.7%, which is similar to the 2017 WHO estimate of diabetes (6.9%). Significant heterogeneity in diabetes and its determinants may exist within Bangladesh due to variations in the level of urbanisation by region, and socioeconomic status of specific subpopulations.[31]

Important determinants of hyperglycaemia in both urban and rural areas of our assessment included hypertension, low physical activity and abdominal obesity. Interestingly, there was no association of diabetes identified for increasing BMI. This indicates that abdominal obesity may be a more significant factor to consider than BMI. Prior studies conducted in Bangladesh have also identified a positive association of central (or abdominal) obesity with diabetes.[32] Furthermore, our assessment found a decrease in prevalence of diabetes with increasing wealth quartile. Prior studies have conflicting findings on the risk of diabetes and other NCDs, among the wealthy based on demographic features such as the area of residence. One prior study conducted in Bangladesh found that people from the highest wealth quintile were more likely to have diabetes than people from the lowest wealth quintile.[33] However, another found a high burden of selected NCDs, including diabetes, among the lowest wealth quintile populations in rural areas and wealthy populations in urban areas.[34] Further study is warranted to assess the reliability of wealth indices as a measurement of socioeconomic status and wealth among Bangladeshi adults.

In our study, a high proportion (~70%) of those with hyperglycaemia self-reported to have been previously diagnosed with diabetes and therefore, aware of their condition. Additionally, 72% reported taking medication to control their diabetes. However, we found that almost one-third of those who self-reported to take medication for their diabetes continued to have hyperglycaemia. Efforts should be made to ensure diabetics in Bangladesh are treated for their condition and secondary prevention of complications of diabetes, such as diabetic retinopathy. This is of particular concern in LMICs where resources are limited and cost-effective solutions for chronic disease treatment should be prioritised. A recently published study found that healthcare expenditure in persons with diabetes in Bangladesh is six times higher than in persons without diabetes.[14] Prevention and management of diabetes are likely to be a cost-saving approach for Bangladesh through the utilisation of community health workers adequately trained to effectively screen for, and identify, people with diabetes.[35]

This study has several strengths. Data collected for our study were of a nationally representative sample indicating our results are generalisable to the population of Bangladeshi adults aged 18 years and above. Additionally, due to our large sample size, we were able to conduct subgroup analyses to identify urban and rural differences. However, several limitations should also be considered when interpreting the results of this analysis. We were unable to measure the prevalence of pre-diabetes and diabetes directly as we did not obtain blood sugar levels using standardised methods, such as fasting blood glucose or 2-hour postprandial measurements. Further, we did not assess each participant's history of classical symptoms of hyperglycaemia, which is necessary to diagnose diabetes according to the ADA guidelines.[21] We were unable to measure known determinants of type 2 diabetes factors such as diet or family history of diabetes. Future studies should consider the addition of glycosylated haemoglobin measurement when assessing the prevalence of diabetes as this method could provide a more long-term and stable

diagnosis of DM. Finally, due to the cross-sectional nature of this study, we were unable to define temporality of certain determinants of hyperglycaemia identified and therefore, unable to assess causality.

## CONCLUSION

Data from this nationally representative sample of Bangladeshi adults aged 18 years and above will be critical to informing the progress of NCD control in Bangladesh per WHO's Global Monitoring Framework and goals for 2025. As our data were collected in late 2015, more recent studies to estimate the prevalence of DM or hyperglycaemia are warranted. Recent changes in risk factor distribution coupled with ageing of the population may have led to changes prevalence of DM not reflected in our results, which will be important in measuring our progress as we approach 2025. We found that about 1 in 20 Bangladeshi adults aged ≥18 years have hyperglycaemia. Among urban residents, we found that about 1 in 10 Bangladeshi adults aged ≥18 years have hyperglycaemia. Bangladeshi adults with hypertension and abdominal obesity are high-risk groups for the development of diabetes and should be targeted for routine screening for diabetes. Preventive methods such as lifestyle changes and medication should be recommended by primary care providers in Bangladesh to avoid the future development of CVDs among this group. In order to control the prevalence of hyperglycaemia in the diabetic range, and reduce the burden of diabetes or associated risk factors, national initiatives such as training community health workers to deliver primary care and implementing universal health coverage should be implemented to curb the spread of NCDs in Bangladesh.

**Acknowledgements** The authors thank Hassanuzzaman Khan for his efforts on data management.

**Contributors** JYI: conceptualised the manuscript, analysed data, interpreted results critically and drafted the manuscript. MMZ: designed the study, interpreted results critically, guided manuscript writing and critically reviewed it. MRB, SAH, SA and ZA-Q: trained the field team, implemented the survey, processed and analysed data and reviewed the manuscript.

**Funding** The study was conducted with the technical and financial assistance of the World Health Organization Country Office for Bangladesh.

**Competing interests** None decared.

**Patient consent for publication** Not required.

**Ethics approval** Ethical guidelines as outlined by the Declaration of Helsinki were followed throughout the study. Ethical clearance was obtained from the Institutional Review Board of Bangabandhu Sheikh Mujib Medical University (BSMMU) (Protocol Number: 1100). We obtained permission from the relevant administrative units of the surveyed districts. Orientations with community leaders (elected representatives of the local government offices) were conducted prior to data collection for community engagement in the study's implementation process.

**Provenance and peer review** Not commissioned; externally peer reviewed.

**Data sharing statement** The deidentified participant data used and/or analysed during the current study are available on reasonable request. Please contact MMZ at zamanm@who.int for further information and guidelines.

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
