## [Reviewer comments · BMJ Open]

ARTICLE DETAILS

TITLE (PROVISIONAL)	Prevalence and determinants of hyperglycemia among adults in Bangladesh: results from a population-based national survey
AUTHORS	Islam, Jessica; Zaman, MM; Bhuiyan, Mahfuz; Haq, Syed; Ahmed, Shamim; Al-Qadir, Zahid

VERSION 1 - REVIEW

REVIEWER	Rachel Climie PARCC, France
REVIEW RETURNED	20-Mar-2019

GENERAL COMMENTS	To the authors; The study by Islam et al. assessed the prevalence and determinants of raised blood glucose among adults in Bangladesh aged ≥ 18 years. The paper is well written and presented. However, I do have some concerns with the study. Please see specific comments below. Biswas et al. (Public health, 2016 Sep;138:4-11) reported prevalence of type 2 diabetes to be 7.4% in Bangladesh in a scoping review in 2016 and in 2017 WHO reported the prevalence of diabetes in adults to be 6.9% in Bangladesh. The current prevalence reported by the authors is much lower than this. An explanation for this is required and it would be useful if the authors could put this into context with regards to prevalence of diabetes in other countries. Further, could the authors provide an explanation as to why prevalence of diabetes is decreasing, if that is the case, in regards to other risk factors for diabetes. For example, the authors showed that abdominal obesity, hypertension and low physical activity were associated with elevated blood glucose levels. Is the prevalence of these risk factors in Bangladesh also changing? What is meant by "raised blood glucose levels"? From the random capillary blood glucose level of ≥ 11.1 mmol/L it indicates presence of diabetes but it would be clearer if the authors could define at first mention that they mean in the diabetic range, rather than just impaired glucose tolerance/prediabetic. It is assumed that the authors are referring to type 2 diabetes, but it would be useful if this was clarified. Are type 1 diabetes cases included?
--

	Did the authors collect data on diet? It is likely that this would be a strong determinant of diabetes. Figure 1 – what is meant by “on study measurement” on the x axis? Do the authors mean “during study measurement”? Figure 2 – please describe in footnote what p value refers to. Page 5 line 122 – “has” should be “have” Please check spelling/formatting throughout as there are minor errors.
--	---

REVIEWER	Manan Pareek North Zealand Hospital, Hilleroed, Denmark Advisory Board and Speaking Honoraria: AstraZeneca; Speaking Honoraria: Bayer and Boehringer Ingelheim
REVIEW RETURNED	20-Apr-2019

GENERAL COMMENTS	Dear Editors, I appreciate the opportunity to serve as a reviewer for this interesting submission. Please find my general and specific comments and suggestions stated below. Islam et al. present a cross-sectional, population-based study on the prevalence and determinants of raised blood glucose among Bangladeshi adults. The work is important and has its merits, but there are also some weaknesses and limitations that need consideration. 1) A few (not too many) grammatical/language errors necessitate thorough review before re-submission. 2) What was the rationale for using the term “raised blood glucose” and not a more conventional term such as “hyperglycemia”? Building on this, why were a “diabetic blood glucose level” and/or use of antidiabetic medications used to define “raised blood glucose”? Even if capillary blood glucose was used, the levels are not far from those obtained using plasma glucose. Basically, the authors are estimating diabetes prevalence, and it might even be underestimated given the poor overlap between subjects diagnosed using different methods. 3) Could some form of validation regarding medication have taken place for at least a subset of patients or was that completely impossible? According to the methods section, treatment history of diabetes was confirmed by prescription, etc., however, it does not look like this was elaborated on in the results section. 4) There are discrepancies between the abstract, the methods section, and the discussion section. According to the methods section, patients were required to have classic symptoms of hyperglycemia. If this is true, the authors should alter “raised blood glucose” to diabetes. However, in the discussion section, it is stated that such symptom history was not assessed. 5) Please make sure to report prevalences and confidence intervals in the same way throughout the text. Do not use different size hyphens, different presence/absence of a space, etc.
---

	6) Strengths and limitations: Please rephrase so that it does not look like individuals aged 18-29 years are the only ones comprising the study population. In the main text, please state whether exclusion of these individuals resulted in prevalence estimates that better reflected those seen in earlier studies of individuals aged at least 30 years from Bangladesh. 7) How do the estimates compare with those from countries with similar economies? 8) Background: Many of the references regarding diabetes prevalence are old. If possible, please update these. 9) Methods: The power calculation belongs in the statistical methods section or at least in a separate, adjacent section. 10) Was the blood pressure measured in the supine or sitting position? 11) I suggest adding numbers to the figures to enhance interpretation.
--	--

VERSION 1 – AUTHOR RESPONSE

Reviewer 1 comments:

Comment 1: Biswas et al. (Public health, 2016 Sep;138:4-11) reported prevalence of type 2 diabetes to be 7.4% in Bangladesh in a scoping review in 2016 and in 2017 WHO reported the prevalence of diabetes in adults to be 6.9% in Bangladesh. The current prevalence reported by the authors is much lower than this. An explanation for this is required and it would be useful if the authors could put this into context with regards to prevalence of diabetes in other countries.

Response: Thank you. We have added the following text to the Discussion to address this comment (page 16, line 397-416):

“Globally, the number of adults living with diabetes has risen from 108 million in 1980 to 422 million in 2013, and low- and middle-income countries (LMICs) have seen the most rapid rise in diabetes prevalence(1). Several lifestyle factors have been attributed to the increase in prevalence across LMICs including, globalization of food production, extensive marketing of low-cost and energy-dense foods, increased sedentary behavior and rapid urbanization(2). In recent decades, the increase in prevalence of diabetes in South Asia has been greater than that seen in high-income countries(3). The prevalence of diabetes in adults across countries in South Asia is similar, excluding Nepal which has a low prevalence in comparison to neighboring countries (8.8% in India, 8.6% in Sri Lanka, 6.9% in Bangladesh, 7.9% in Bhutan, 6.9% in Pakistan, and 4.0% in Nepal)(4). In our study, we observed a prevalence of 5.6% hyperglycemia in the diabetic range among adults aged 18 years and above. This prevalence is lower than previous studies conducted in Bangladesh and neighboring countries. In fact, a scoping review estimated the pooled prevalence of type-2 diabetes to be 7.4%(5). Our estimated prevalence may be lower due to a younger study population (18 years and above compared to the WHO estimate among adults aged 30 years and above), and a higher percentage of participants from rural areas (61.1%, as is representative of Bangladesh). Indeed, when we restrict our analytic sample to 30 years and above, the prevalence of hyperglycemia is 6.7%, which is similar to the 2017 WHO estimate of diabetes (6.9%). Therefore, between studies comparison should not be done without considering the sample characteristics and other methodological differences. Significant

heterogeneity in diabetes and its determinants may exist within one country due to variations in level of urbanization by state, ethnic phenotypes, and socioeconomic status of specific sub-populations(6) etc.”

Comment 2: Further, could the authors provide an explanation as to why prevalence of diabetes is decreasing, if that is the case, in regards to other risk factors for diabetes. For example, the authors showed that abdominal obesity, hypertension and low physical activity were associated with elevated blood glucose levels. Is the prevalence of these risk factors in Bangladesh also changing?

Response: We do not believe the prevalence of diabetes is decreasing in Bangladesh. Our study population was generally younger (18 years and below), compared to prior studies (mostly 30 years and above). Additionally, to reflect the population distribution of Bangladesh, 61% of participants in our study resided in rural areas, which has been previously documented to have lower prevalence of diabetes than urban areas. Please see the addition to the discussion shown in response to Comment 1.

Comment: 3 What is meant by “raised blood glucose levels”? From the random capillary blood glucose level of ≥ 11.1 mmol/L it indicates presence of diabetes but it would be clearer if the authors could define at first mention that they mean in the diabetic range, rather than just impaired glucose tolerance/prediabetic.

Response: Thank you for your comment. We have replaced the phrase “raised blood glucose levels” to hyperglycemia throughout the manuscript. Additionally, we have updated the primary outcome measure definition as follows (page 2, line 47 & page 9, line 217-220): “Hyperglycemia was defined as a random capillary blood glucose level of ≥ 11.1 mmol/L (i.e. in the diabetic range) or currently taking medication to control diabetes, based on self-report.”

Comment 4: It is assumed that the authors are referring to type 2 diabetes, but it would be useful if this was clarified. Are type 1 diabetes cases included?

Response: Yes, we are referring to type-2 diabetes. We asked participants of their history of chronic disease, including type-1 diabetes. None of the participants self-reported to have type-1 diabetes. We have included the following in the “Background characteristics” of the results section (page 12, line 309): “Additionally, no participants reported to have been previously diagnosed with type-1 diabetes.”

Comment 5: Did the authors collect data on diet? It is likely that this would be a strong determinant of diabetes.

Response: We agree that diet is a strong determinant of diabetes. However, we did not collect data on diet. We have updated the limitations section of the Discussion as follows (page 18, line 462): “Additionally, we were unable to measure known determinants of type-2 diabetes such as diet or family history of diabetes.”

Comment 6: Figure 1 – what is meant by “on study measurement” on the x axis? Do the authors mean “during study measurement”?

Response: The x-axis has been updated to read: “Blood glucose levels (mmol/L).”

Comment 7: Figure 2 – please describe in footnote what p value refers to.

Response: The following footnote has been added to Figure 2: “Exact test p-value to assess the relationship between prevalence of raised blood glucose and age group stratified by sex.”

Comment 8: Page 5 line 122 – “has” should be “have”

Response: Thank you. The text has been updated accordingly.

Comment 9: Please check spelling/formatting throughout as there are minor errors.

Response: We have updated the manuscript to correct for spelling and formatting errors.

Reviewer 2: Comments

Comment 10: A few (not too many) grammatical/language errors necessitate thorough review before re-submission.

Response: Thank you. We have reviewed the manuscript and updated the text for grammatical errors.

Comment 11: What was the rationale for using the term “raised blood glucose” and not a more conventional term such as “hyperglycemia”? Building on this, why were a “diabetic blood glucose level” and/or use of antidiabetic medications used to define “raised blood glucose”? Even if capillary blood glucose was used, the levels are not far from those obtained using plasma glucose. Basically, the authors are estimating diabetes prevalence, and it might even be underestimated given the poor overlap between subjects diagnosed using different methods.

Response: Thank you for your comment. Reviewer 1 raised a similar concern. We have replaced the phrase “raised blood glucose” with hyperglycemia as recommended. In addition, please see response to Comment 3.

Comment 12: Could some form of validation regarding medication have taken place for at least a subset of patients or was that completely impossible? According to the methods section, treatment history of diabetes was confirmed by prescription, etc., however, it does not look like this was elaborated on in the results section.

Response: Treatment history was confirmed using medicine strips or insulin vials, or prescription records. We have updated the Strengths and Limitations section to state the following (page 4, line 103): “We assessed diabetes medication history based on self-report and we were able to obtain medicine strips or vials or prescription records of participants to confirm the self-reported data.” Additionally, we have included the following in the Results section (page 13, line 351): “We were able to confirm 100% of participant’s self-reported diabetes treatment history by checking prescriptions or medicine strips/vials.”

Additionally, we have added to Table 1 the proportion of participants who self-reported to take diabetes medication. This has also been summarized in the Results (Page 14, line 353).

Comment 13: There are discrepancies between the abstract, the methods section, and the discussion section. According to the methods section, patients were required to have classic symptoms of hyperglycemia. If this is true, the authors should alter “raised blood glucose” to diabetes. However, in the discussion section, it is stated that such symptom history was not assessed.

Response: Thank you for pointing this out. We have updated the Methods section to exclude the phrase “with classic symptoms of hyperglycemia.” The definition of our outcome is now as follows (page 8, line 217-220): “An individual was considered to have hyperglycemia if the plasma glucose level was 11.1 mmol/L or higher (i.e. in the diabetic range), and/or if they self-reported to take diabetes medication.” The definition is consistent across the abstract and methods section. The limitation pointed out in the Discussion section is accurate.

Comment 14: Please make sure to report prevalences and confidence intervals in the same way throughout the text. Do not use different size hyphens, different presence/absence of a space, etc.

Response: We have gone through and revised the manuscript text to ensure prevalence and confidence intervals reporting style is consistent.

Comment 15: Strengths and limitations: Please rephrase so that it does not look like individuals aged 18-29 years are the only ones comprising the study population. In the main text, please state whether

exclusion of these individuals resulted in prevalence estimates that better reflected those seen in earlier studies of individuals aged at least 30 years from Bangladesh.

Response: We have updated the Strength as follows (page 4, line 90): “•Currently, nationally representative data for the prevalence of hyperglycemia in the diabetic range is unavailable for adults aged 18-29 years. A strength of our study is that we included Bangladeshi adults aged 18 and above to obtain novel data on the prevalence of hyperglycemia, diabetes and relevant non-communicable disease risk factors.

We have included the following sentence in the Discussion as described in the response to Comment 1: “Indeed, when we restrict our analytic sample to 30 years and above, the prevalence of hyperglycemia is 6.7%, which is similar to the 2017 WHO estimate of diabetes (6.9%).”

Comment 16: How do the estimates compare with those from countries with similar economies?

Response: We have included the following sentence in the Discussion (page 16, line 405-408): “The prevalence of diabetes in adults across countries in South Asia is similar, excluding Nepal which has a low prevalence in comparison to neighboring countries (8.8% in India, 8.6% in Sri Lanka, 6.9% in Bangladesh, 7.9% in Bhutan, 6.9% in Pakistan, and 4.0% in Nepal)(4).” For further details, please see the added paragraph included in response to Comment 1.

Comment 17: Background: Many of the references regarding diabetes prevalence are old. If possible, please update these.

Response: Thank you for your comment. We have removed the old references [(Reference 1: Lozano R et al. 2010), (Reference 2: Murray CJ et al, 2010), (Reference 4: Wild S et al. 2004)] and updated the References to include more up to date sources including:

Reference 1. Disease GBD, Injury I, Prevalence C. Global, regional, and national incidence, prevalence, and years lived with disability for 354 diseases and injuries for 195 countries and territories, 1990-2017: a systematic analysis for the Global Burden of Disease Study 2017. *Lancet* (London, England) 2018;392(10159):1789-858. doi: 10.1016/S0140-6736(18)32279-7

Reference 2. Ogurtsova K, da Rocha Fernandes JD, Huang Y, et al. IDF Diabetes Atlas: Global estimates for the prevalence of diabetes for 2015 and 2040. *Diabetes research and clinical practice* 2017;128:40-50. doi: 10.1016/j.diabres.2017.03.024

Reference 4. Guariguata L, Whiting DR, Hambleton I, et al. Global estimates of diabetes prevalence for 2013 and projections for 2035. *Diabetes research and clinical practice* 2014;103(2):137-49. doi: 10.1016/j.diabres.2013.11.002

Comment 18: Methods: The power calculation belongs in the statistical methods section or at least in a separate, adjacent section.

Response: We have updated the Methods section and designated a separate section titled (Page 7, line 169) "Sample size estimation" for the paragraph describing the power calculations.

Comment 19: Was the blood pressure measured in the supine or sitting position?

Response: We have updated the methods to include the following sentence: "Blood pressure measurements were consistently taken on each participant's right arm at the level of the heart and elbow-assisted, while the participant was in a seated position."

Comment 20: I suggest adding numbers to the figures to enhance interpretation.

Response: We have updated Figures 2 – 4 to include the plotted values. We did not update Figure 1 to include plotted values as the figure includes box plots.

1. Global report on diabetes. Geneva, Switzerland: World Health Organization 2016.
2. Singh PN, Arthur KN, Orlich MJ, James W, Purty A, Job JS, et al. Global epidemiology of obesity, vegetarian dietary patterns, and noncommunicable disease in Asian Indians. *The American journal of clinical nutrition*. 2014;100 Suppl 1:359S-64S. doi: 10.3945/ajcn.113.071571. PubMed PMID: 24847857; PubMed Central PMCID: PMC4144108.
3. Jayawardena R, Ranasinghe P, Byrne NM, Soares MJ, Katulanda P, Hills AP. Prevalence and trends of the diabetes epidemic in South Asia: a systematic review and meta-analysis. *BMC public health*. 2012;12:380. doi: 10.1186/1471-2458-12-380. PubMed PMID: 22630043; PubMed Central PMCID: PMC3447674.
4. Hills AP, Arena R, Khunti K, Yajnik CS, Jayawardena R, Henry CJ, et al. Epidemiology and determinants of type 2 diabetes in south Asia. *Lancet Diabetes Endocrinol*. 2018;6(12):966-78. doi: 10.1016/S2213-8587(18)30204-3. PubMed PMID: 30287102.
5. Biswas T, Islam A, Rawal LB, Islam SM. Increasing prevalence of diabetes in Bangladesh: a scoping review. *Public health*. 2016;138:4-11. doi: 10.1016/j.puhe.2016.03.025. PubMed PMID: 27169347.

6. Chan JC, Malik V, Jia W, Kadowaki T, Yajnik CS, Yoon KH, et al. Diabetes in Asia: epidemiology, risk factors, and pathophysiology. *Jama*. 2009;301(20):2129-40. doi: 10.1001/jama.2009.726. PubMed PMID: 19470990.
1. Global report on diabetes. Geneva, Switzerland: World Health Organization 2016.
 2. Singh PN, Arthur KN, Orlich MJ, James W, Purty A, Job JS, et al. Global epidemiology of obesity, vegetarian dietary patterns, and noncommunicable disease in Asian Indians. *The American journal of clinical nutrition*. 2014;100 Suppl 1:359S-64S. doi: 10.3945/ajcn.113.071571. PubMed PMID: 24847857; PubMed Central PMCID: PMC4144108.
 3. Jayawardena R, Ranasinghe P, Byrne NM, Soares MJ, Katulanda P, Hills AP. Prevalence and trends of the diabetes epidemic in South Asia: a systematic review and meta-analysis. *BMC public health*. 2012;12:380. doi: 10.1186/1471-2458-12-380. PubMed PMID: 22630043; PubMed Central PMCID: PMC3447674.
 4. Hills AP, Arena R, Khunti K, Yajnik CS, Jayawardena R, Henry CJ, et al. Epidemiology and determinants of type 2 diabetes in south Asia. *Lancet Diabetes Endocrinol*. 2018;6(12):966-78. doi: 10.1016/S2213-8587(18)30204-3. PubMed PMID: 30287102.
 5. Biswas T, Islam A, Rawal LB, Islam SM. Increasing prevalence of diabetes in Bangladesh: a scoping review. *Public health*. 2016;138:4-11. doi: 10.1016/j.puhe.2016.03.025. PubMed PMID: 27169347.
 6. Chan JC, Malik V, Jia W, Kadowaki T, Yajnik CS, Yoon KH, et al. Diabetes in Asia: epidemiology, risk factors, and pathophysiology. *Jama*. 2009;301(20):2129-40. doi: 10.1001/jama.2009.726. PubMed PMID: 19470990.

VERSION 2 – REVIEW

REVIEWER	Rachel Climie INSERM U970, France
REVIEW RETURNED	27-May-2019

GENERAL COMMENTS	The authors have addressed my earlier comments and the manuscript is now acceptable for publication.
--

REVIEWER	Manan Pareek Department of Cardiology, North Zealand Hospital, Hillerød, Denmark Advisory Board and Speaking Honoraria: AstraZeneca; Speaking Honoraria: Bayer and Boehringer Ingelheim.
REVIEW RETURNED	27-May-2019

GENERAL COMMENTS	The authors have done an excellent job revising this paper.
---